# Use of Hyperspectral Imaging for the Quantification of Organic Contaminants on Copper Surfaces for Electronic Applications

**DOI:** 10.3390/s21165595

**Published:** 2021-08-19

**Authors:** Tim Englert, Florian Gruber, Jan Stiedl, Simon Green, Timo Jacob, Karsten Rebner, Wulf Grählert

**Affiliations:** 1Robert Bosch GmbH, Automotive Electronics, Postfach 1342, 72703 Reutlingen, Germany; Tim.Englert2@de.bosch.com (T.E.); Jan.Stiedl@de.bosch.com (J.S.); Simon.Green@de.bosch.com (S.G.); 2Institute of Electrochemistry, Ulm University, Albert-Einstein-Allee 47, 89081 Ulm, Germany; timo.jacob@uni-ulm.de; 3Fraunhofer Institute for Material and Beam Technology IWS, Winterbergstraße 28, 01277 Dresden, Germany; wulf.graehlert@iws.fraunhofer.de; 4Process Analysis and Technology PA & T, Reutlingen University, Alteburgstraße 150, 72762 Reutlingen, Germany; karsten.rebner@reutlingen-university.de

**Keywords:** spectral imaging, HSI, XPS, AES, multivariate analysis, machine learning, elastic net, RF, SVM, organic residues, cleaning after soldering, cleanliness

## Abstract

To correctly assess the cleanliness of technical surfaces in a production process, corresponding online monitoring systems must provide sufficient data. A promising method for fast, large-area, and non-contact monitoring is hyperspectral imaging (HSI), which was used in this paper for the detection and quantification of organic surface contaminations. Depending on the cleaning parameter constellation, different levels of organic residues remained on the surface. Afterwards, the cleanliness was determined by the carbon content in the atom percent on the sample surfaces, characterized by XPS and AES. The HSI data and the XPS measurements were correlated, using machine learning methods, to generate a predictive model for the carbon content of the surface. The regression algorithms elastic net, random forest regression, and support vector machine regression were used. Overall, the developed method was able to quantify organic contaminations on technical surfaces. The best regression model found was a random forest model, which achieved an R^2^ of 0.7 and an RMSE of 7.65 At.-% C. Due to the easy-to-use measurement and the fast evaluation by machine learning, the method seems suitable for an online monitoring system. However, the results also show that further experiments are necessary to improve the quality of the prediction models.

## 1. Introduction

What does it mean to be a clean surface and how can it be measured? The investigation of cleanliness is becoming more of a focus in applied research. The challenge is that more applications and processes are moving toward the need for higher levels of cleanliness. Therefore, the limit of the required cleanliness must be defined and measurable. In order to determine the cleanliness of a surface with regard to a specific contamination, a suitable method must be identified, and this method must also meet application-specific requirements such as spatial resolution and measurement speed. Especially for electronic production, applications with a higher voltage or current in relation to a smaller product size have higher cleanliness requirements. An insufficient level of cleanliness leads to electrochemical migration, oxidation, and a decrease in adhesion, which reduces reliability [1,2,3].

In the production of electronics for the automotive industry, the reliability of the car components is a key requirement. A lot of research has been to investigate the reliability for direct packing, gluing, or wire bonding. It was shown that oxides and contaminations on the surface were linked to an increase of or a decrease in the adhesion of processes such as wire bonding, gluing, or molding. For oxides, the adhesion increases until the oxide reaches a critical thickness [4,5,6,7,8]. Lower organic contamination leads to the improved adhesion of these processes. On the other hand, a higher amount of organic contamination on the surface reduces adhesion and leads to a decrease in reliability [9,10,11,12]. Quantifying the level of cleanliness, depending on process and applications, is important in order to control the reliability of the product.

For quantifying organic residues, chemical analytics and a knowledge of the properties of the contaminations is necessary. During the soldering process, organic contamination is mainly caused by the use of fluxes in the solder pastes. For a soldering process, it is required that the liquid solder can flow and wet the surface. Therefore, solder pastes contain flux. These fluxes are primarily based on rosin. Rosin is a natural product containing a high proportion of organic acids. The primary acids of the rosin are abietic acid, succinic acid, and glutaric acid [13,14,15].

Additionally, the fluxes contain high amounts of alcohols. The alcohols volatilize and decompose during the soldering process. Consequently, the residues on the surface consist of organic acids, which present as a salt. In the literature, the corrosive properties of these salt residues and their impact on the reliability of electronics in humid environments is reported [16,17,18,19].

For cleanliness quality control, visual inspection by a trained operator is a state-of-the-art process in industrial serial production. A fast, non-destructive, objective, and online capable cleanliness measurement would be highly beneficial for costs and quality in a manufacturing process. A monitoring system can quantify the level of cleanliness and help to objectively compare the performance of the cleaning process with alternative cleaning processes. Therefore, the best performing and most cost-efficient process for this application needs to be identified [20].

To measure the cleanliness of metal surfaces, X-ray photoelectron spectroscopy (XPS) measurements are described in the literature. The level of cleanliness for organic contaminations is determined by the carbon content on the surface in atom percent (At.-%) [21]. In addition to XPS, Auger electron spectroscopy (AES), and time-of-flight-secondary ion mass spectroscopy (TOF-SIMS) are well known to characterize organic contaminations on metal substrates [22,23]. However, because of the required ultra-high vacuum and the limited dimensions of the sample size, these methods are not options for an online monitoring process [24]. Techniques such as contact angle measurement, ion exchange chromatography, and spectroscopy in the ultraviolet and visible spectral range (UV-Vis) are described in the literature for the investigation of organic substances [24,25,26,27]. The disadvantage of these methods is that sample preparation and a trained employee is needed and that they are limited to single point measurements. Consequently, they are limited to at-line measurements, and a large-area inline measurement is not possible. In addition, the methods require a liquid that interacts with the surface for contact angle measurement and ion exchange chromatography. Depending on the liquid, the liquid may possibly change the surface properties and the adhesion of subsequent processes [24].

Diffuse UV-Vis reflection spectroscopy measures the absorption and the scattering of the sample surface without making contact with the surface, and it is commonly used for quantitative and qualitative analysis [28,29]. In the literature, examples of an online monitoring method can be found, for example, in powder analysis applications [30] or in the food industry [31]. 

However, such a monitoring system can only be used for a small portion of the surface. The UV-VIS diffuse reflection method simply cannot be used for larger and geometrically more complex surfaces due to its protruding components. 

Another challenge for all of the methods mentioned above is whether the cleanliness at the measured point is representative of the entire surface. In real applications or processes, inhomogeneously contaminated surfaces often occur. It can be concluded that a measurement at one point on the surface is not sufficient to determine the cleanliness of the entire surface. As a result, imaging methods that measure the entire surface have a significant advantage over conventional methods. Hyperspectral imaging (HSI) has the potential to meet the requirements for high resolution and large area detection and assessment of surface contaminations. HSI enables the rapid spatially resolved spectral analysis of surfaces. HSI systems are well described in the literature for applications in the fields of medicine, food, and agriculture [32,33,34]. Paoletti et al. [35] provide a comprehensive review of the current state-of-the-art of methods for HSI classification, analyzing, and correlation of imaging data. To monitor cleanliness, some approaches for the food industry have been published using a HSI system [36,37,38]. Various approaches have been published for the quantification of thin films, e.g., oxides or liquid films [39,40,41,42]. Stiedl et al. [40] have shown that the prediction of oxide layer thickness on technical copper using HSI and multivariate analysis is possible.

In this research, hyperspectral imaging will be used to evaluate the cleanliness of surfaces of soldered copper substrates and the degree of contamination by the organic contaminants on these surfaces. The hyperspectral imaging measurements were conducted in the visible and near-infrared (VNIR) spectral range.

The actual level of contamination is determined by the carbon content on the sample surfaces as measured by XPS and AES. Different levels of surface cleanliness were generated by different cleaning process parameters. The cleanliness of the surfaces was quantified by the carbon content in atom percent. The HSI data and the carbon content measurements were correlated using machine learning methods to generate a predictive model for the carbon content of the surface.

## 2. Materials and Methods

### 2.1. Sample Preparation

Direct bonded copper (DBC) Curamik^®^Power substrates (Rogers Corporation, Chandler, AZ, USA) with the dimensions of 21.0 mm × 21.0 mm × 1.1 mm were used. The solder paste F360 SnAg 3.5 (Hereaus, Hanau, Germany) was printed with the dimensions of 18.0 mm × 3.0 mm × 0.2 mm on the surface. The samples were soldered at 240 °C for 4 min in a nitrogen atmosphere, cooled down to 25 °C, and directly cleaned. For each set of samples (Set1–Set7), 10 DBC substrates were soldered and cleaned afterwards. Figure 1 shows a photo of some of the soldered and cleaned samples.

### 2.2. Wet Chemical Cleaning Process

All samples were cleaned with a mixture of the cleaning agents Vigon A200 (Zestron, Ingolstadt, Germany) and CI20 (Zestron, Ingolstadt, Germany) and deionized water. The cleaning process parameters were the temperature of the cleaning agent, the duration of the cleaning process, the concentration of A200 and CI20, and the rinsing duration with deionized water after the cleaning. These parameters were varied to produce different levels of cleanliness. The cleaning parameters are shown in Table 1 for each set of cleaning parameters (Set1–Set7). The rinsed samples were dried using compressed dry nitrogen, were wrapped up in aluminum foil, and were packed in a polymer bag with a nitrogen gas atmosphere to prevent recontamination. 

### 2.3. X-ray Photoelectron Spectroscopy (XPS)

The measurements were conducted under the following experimental conditions: A system base pressure of 4.0 × 10^−10^ mbar was used. A monochromatic Al Kα radiation was used, and the anode tube operated at 12.5 kV with 20 mA. The take-off angle for the electrons was 0° with respect to the normal surface. The XPS core level spectra were measured with standard X-ray source SPECS XR50 (SPECS Surface Nano Analysis GmbH, Berlin, Germany) and the concentric hemispherical analyzer Phoibos 100, SPECS (SPECS Surface Nano Analysis GmbH, Berlin, Germany). The pass energy of the concentric hemispherical analyzer was 50 eV for the survey and 20 eV for the high-resolution spectra. The data acquisition was 0.5 eV; 0.1 eV per step. For each sample the measurements were performed in the same spot on the substrate surface. Spot A was defined at 1 mm away from solder paste on the copper, and Spot B was in the center of the substrate on the copper. The XPS measurement was the spot size, which was defined of 100 µm × 100 µm.

### 2.4. Auger Electron Spectroscopy (AES)

The measurements were conducted using an electron beam voltage of 3.0 keV. The chamber pressure was set at 3 × 10^−8^ mbar during measurements. A Perkin Elmer PHI-600 scanning Auger spectrometer (PerkinElmer, Waltham, MA, USA) equipped with a Perkin Elmer 04-303 differential ion gun (PerkinElmer, Waltham, MA, USA) was used. The AES line scans of the inhomogeneous distribution of the carbon surface contamination were started about 1 mm away from solder paste boundaries on the sample.

### 2.5. Hyperspectral Imaging (HSI)

The hyperspectral measurement of the samples was performed with a pushbroom HSI measuring system with diffuse halogen illumination. A schematic representation of the HIS system and a photo of the HIS system with the diffuse illumination is shown in Figure 2. The system was equipped with a Specim FX10 VNIR HSI camera (Specim Spectral Imaging Ltd., Oulu, Finland) with a wavelength range between 400 nm and 1000 nm and a matching lens (f/1.7, FOV 38°). The HSI camera was equipped with a CMOS detector with 1024 pixel in the spatial dimension and 224 pixels in the spectral dimension, and no binning was used. Lighting was provided by 6 halogen lamps with a power of 25 W each. The diffuse illumination of the samples was achieved by a self-designed integration tube made of optical PTFE (Spectralon, Labsphere Inc., North Sutton, NH, USA). This produces a diffuse illumination without a specific angle of incidence. The movement of the samples was controlled by a linear stage (VT 80, PI Micos, Eschbach, Germany). The control of the system components and the data acquisition was conducted using the dedicated HSI software suite imanto^®^pro (Fraunhofer IWS, Dresden, Germany).

To avoid irregularities in the lighting and to eliminate the influence of dark current, a white and a dark correction for each wavelength according to Equation (1) was conducted.
(1)$$I_{c}\left(\lambda \right)=\frac{I_{o}\left(\lambda \right)-I_{d}\left(\lambda \right)}{I_{w}\left(\lambda \right)-I_{d}\left(\lambda \right)}$$

I_c_ is the corrected image, and Io is the original image for wavelength λ. I_d_ is the dark signal recorded with the light source switched off and the lens covered, and I_w_ is the white reference. For the white reference, an optical PTFE plate was recorded under the same measuring conditions as the original image.

The measurements of the samples from all of sets were performed with a working distance of 300 mm, an exposure time of 20 ms, and a recording frame rate of 50 Hz. This resulted in a field of view (FOV) of ~150 mm, a spatial resolution of ~150 μm, and a spectral resolution of ~5.5 nm. The speed of the linear stage was set to 4 mm s^−1^ to obtain squared pixels. The result of each measurement was a hypercube with 224 spectral bands between 400 nm and 1000 nm. 

The acquisition and basic preprocessing of the hyperspectral data was achieved by using the imanto^®^pro software package (Fraunhofer IWS Dresden, Dresden, Germany). This software packaged allowed us the control the measurement parameters, the visualization, and pre-processing of the acquired hyperspectral images, which allowed modifications to the images such as smoothing or normalization.

### 2.6. Data Analysis and Machine Learning

The obtained hyperspectral data and the ground truth data for the carbon content were used to train the prediction models. The goal was to train machine learning models to predict the carbon content at each point of the sample based on its spectral information. There were three different regression algorithms that were trained and compared—ElasticNet regression (EN [43]), random forest regression (RF, [44]), and support vector machine regression (SVM, [45]). For a detailed description of the algorithms, please refer to the corresponding literature. Before model training, an optional normalization using vector normalization (VN) or standard normal variate correction (SNV), an optional principal component analysis (PCA), an optional standardization to a standard deviation of one, and a mean value of zero was performed.

For all of the algorithms an optimization of their hyperparameters was conducted. Hyperparameters are the parameters of algorithms that are used to control the algorithms themselves. These parameters are set before the actual training of the models and are not learned during the training. The hyperparameters can have a large effect on the overall classification accuracy of the trained algorithms, which is why an optimization is useful. An example of an algorithm for the automated hyperparameter optimization is the Bayesian optimization algorithm (BOA, [46]). A disadvantage of this optimization procedure is that running the algorithm itself is time consuming and computationally intensive. A quick and easy alternative is the random search algorithm (RS), which randomly selects the next hyperparameter of the optimization routine. It has been shown that the differences between the sets of the hyperparameters found by RS and other hyperparameter optimization algorithms are often small [47]. Therefore, the RS algorithm is used here for hyperparameter optimization. The optimized hyperparameters and the ranges in which these parameters were optimized are summarized in Table A1. All of the other parameters were left at their default settings. Hyperparameter optimization using RS was performed on 30 epochs, which means that 30 models with different random parameters were trained to find a good set of hyperparameters. 

For the training of the prediction model, the full dataset was first split into a training set containing 70% of the data and a test set containing 30% of the data. The training dataset was used to optimize the hyperparameters. Therefore, a 5-fold cross validation was used, and the root mean squared error (RMSE) was determined. The hyperparameter combination, which led to the lowest RMSE of cross validation, was selected, and a final model was trained using all of the training data. Thereafter, this model was used to predict the data in the test set, and the RMSE of prediction and the correlation coefficient R^2^ was calculated. 

All of the calculations were performed with scikit-learn (version 0.23.2., [48]) and a Windows 10 computer with an Intel CoreTM i5-4590 with 3.3 GHz, 16 GB RAM and a Nvidia GTX 1080 Ti graphics card with 11 GB GDDR5X memory and a processor clock of 1632 MHz.

## 3. Results and Discussion

### 3.1. XPS and AES Measurements

From the sample surface, counts for Cu2p_1/2_, Cu2p_3/2_, C1s, O1s, Sn3d_5/2_, and Ag3d_5/2_ were collected by XPS. The copper signal is from the sample surface. The oxygen signal is from oxides and organics. The carbon signal is from the organic residues of the flux. Tin and silver are components of the solder paste. The carbon content in atomic percent (At.-%) was calculated through the relative counts divided by the photoionization factors described by the literature [49].

The carbon content from Set1 to Set7 is presented in Table 1. The variations in the carbon were between 36–79 At.-%, depending on the measurement spot and cleaning process parameters.

Set1 showed the highest level of carbon at 76 At.-% near the solder paste and 46 At.-% in the center of the sample. This could be explained by Set1 having the shortest cleaning duration and the lowest concentration of the cleaning agents A200 and CI20. For Set5, the measured level of carbon was 63 At.-% near the solder paste and 40 At.-% in the center. The lower carbon content is caused by the higher cleaning temperature and the higher cleaning agent concentration. Sets6 and 7 were cleaned using the same parameters, resulting in the lowest measured carbon contents and therefore the best level of cleanliness.

In Figure 3, the XPS survey spectra and C1s spectra of Set1, Set5, and Set7 are shown. These are the sets with highest, the lowest, and an average amount of carbon measured at Spot A. The decrease of the C1s counts at 285 eV for Set7 compared to Set1 is visible. Furthermore, in the C1s spectra of Set1, the carboxylic acid (COOR) peak is the most pronounced. This can be explained by the higher proportion of organic salts and acids due to insufficient cleaning.

In Figure 4, the AES spectra of the contaminant peaks of Set4 is presented. The carbon counts were taken from the C KVV Auger peak at 269.5 eV, and the oxygen counts were taken from the O KVV peak at 514.5 eV. In addition, a Sn MNN peak was found in the AES spectrum [50,51]. The AES measurements confirm that there are organic residues on the surface. An AES line scan was used to investigate the distribution of these organic residues. The AES line scan was started on the copper surface about 1 mm away from the solder paste and was directed towards the center of the sample. The closer the measurement spot was taken to the center of the sample, the farther away it was from the solder paste, and thus, the further away it was from the origin of the organic residues. This effect was directly reflected in the general decrease of the carbon signal intensity in the AES line profile in Figure 4. Local inhomogeneity in the carbon contamination can be recognized by irregularities in the intensity development of the line scan. The oxygen counts were only observed at the first 250 µm. One can see that the contamination is higher in the first 250 µm of the line scan and is chemically different from the rest. One possible hypothesis would be that the amount of organics on the surface depends on the geometric distance to the solder paste boundary. The closer it is measured to the solder paste boundary, the higher the contamination was.

The XPS and AES measurements show that the amount of organic residues on the surface differs depending on the cleaning process parameters and the distance to the solder paste. It was shown that a variation of the cleaning process parameters leads to different amounts flux residues on the surface.

### 3.2. HSI Data Evaluation and Modeling

Figure 5 shows an example HSI measurement of the samples from Set2. The example spectra show the absorption of the copper at wavelengths below 600 nm. The increase in reflectivity at wavelengths below 450 nm is caused by stray light within the camera. In both the spectra and in the image of the color-coded reflectivity, the differences between the less heavily contaminated areas in the centre of the Cu samples (green crosses/spectra) and the heavily contaminated areas near the soldered areas (red crosses/spectra) can be seen. A stronger contamination leads to a decrease of the reflected intensity and a change in the shape of the spectra, which is caused by absorption and scattering by the contaminants.

To obtain data that can be used to train regression models to predict the organic contamination on the samples, spectra from each sample of each sample set were selected manually. For each sample set, two XPS measurements of the carbon content were performed (Spot A and B). Therefore, the spectra were also taken from those two areas for each sample (see Figure 6). From each sample set and from each of the two spots, A and B, 100 spectra were selected at random, resulting in a total of 1400 spectra for all samples from the HSI hypercubes. These spectra can be assigned to 14 different carbon contamination levels. Consequently, there are considerably more spectra than ground truth values. The obtained spectra and the ground truth values of the carbon contamination can then used for the training, optimization, and validation of regression models.

Figure 7 shows the mean spectra for 5 of the 14 carbon contamination values. One can see a clear trend in the spectra correlated with the amount of carbon contamination. Higher contamination leads to a lower overall reflectivity and a change in the shape of the spectra. Besides this, the spectra are typical for copper substrates, and there are no additional bands that are visible. Based on these results, the prediction of carbon contamination based on the hyperspectral data seems possible.

With the ground truth data of the carbon content and the spectra from the HSI measurements, the regression models for the prediction of the carbon content were trained. Table 2 shows the RMSE of prediction and the R^2^ of the best-found models after hyperparameter optimization for each of the three algorithms. The best results with a RMSE of prediction of 7.65 At.-% C and a R^2^ of 0.7were achieved for the random forest model. 

Figure 8 shows the prediction results of the best-found RF model. The figure and the RMSE and R^2^ values obtained show that there is a correlation between the hyperspectral measurements and the measured value of the XPS, i.e., the amount of carbon in At.-%. However, it can also be seen that there are sometimes considerable differences within the spectra assigned to a carbon loading, which also lead to large differences in the predicted carbon content. This can be explained by the fact that only two XPS measurements were conducted for each sample set, but several areas were selected as training data in the hyperspectral measurements. It can therefore be assumed that the carbon content within the selected areas is subject to greater fluctuations that are not covered by the XPS measurements. These fluctuations in the carbon contamination were observed by the AES measurements. In order to improve the accuracy of the obtained models, further XPS measurements of the carbon loading should be conducted, and the HSI data for model training should only be selected in the actual measurement range of the XPS measurements.

Figure 9 shows the colour coded prediction results for the carbon content in atomic percent for the samples from Set2, calculated using the best-found RF model. For some areas (marked with a box), the mean value of the predicted carbon content is shown, and there is a relatively good agreement in the ground truth carbon content values measured using XPS. The results also show that there are relatively large differences in the calculated carbon content of each sample, especially in the strongly contaminated areas next to the soldered regions. It is assumed that these differences are real and were not captured by the XPS measurements at only one point. As already described, this can explain the large variation in the prediction of carbon content by the regression models.

Nevertheless, the results show that hyperspectral imaging in combination with machine learning is able to predict organic contamination and carbon loading on soldered copper surfaces. Due to the fast and non-contact measurement method and the fast evaluation by artificial intelligence, the developed method seems to be very well suited for the online monitoring of the cleanliness of soldered copper samples after they have undergone a cleaning process. Before practical implementation, however, further experiments have to be conducted to increase the data basis. In particular, more ground truth XPS measurements are needed to increase the quality of the prediction models and to have independent data for the validation of the models.

## 4. Conclusions

It was shown that the cleanliness of soldered copper substrates depended on the cleaning process parameters. These parameters determine the level of cleanliness, which can be defined as the carbon content in atomic percent (At.-%) measured by XPS. Additionally, one could see through the AES and HSI data that these carbon contaminations were inhomogeneously distributed on the surface. A correlation between the spectra and the amount of carbon contamination was found. A higher organic contamination leads to a lower overall reflectivity. With the XPS and HSI measurements, the regression models for the prediction of the carbon content were trained. The random forest regression (RF) was the best found method, with a RMSE of 7.65 At.-% C. To further improve the accuracy of the model, XPS measurements are necessary. The predicted values of the presented Set2 were in good agreement with the XPS measurements. Overall, the developed method seems to be suited for non-contact and large-area inline monitoring to quantify organic contamination on technical surfaces. However, the results also show that more data are necessary to obtain a more reliable statement and to transfer the presented principle into a technical implementation. In particular, more ground truth values (XPS measurements) are necessary to increase the quality of the prediction models and to obtain more independent test data for the validation of the results.

## Figures and Tables

**Figure 1 sensors-21-05595-f001:**
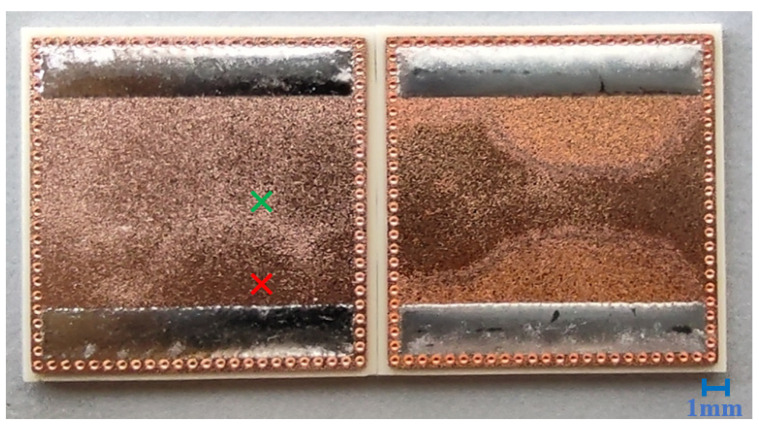
Photo of some of the soldered and cleaned DBC samples. The left-hand side shows Set7 after cleaning, and the right-hand side shows it before cleaning. The red point marks measuring Spot A, and the green point marks measuring Spot B.

**Figure 2 sensors-21-05595-f002:**
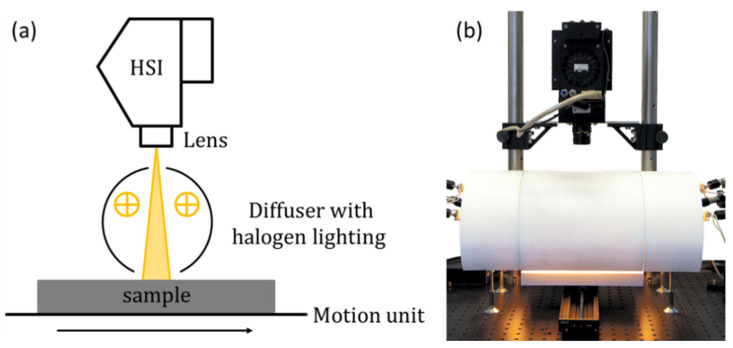
(**a**) Schematic representation of the hyperspectral imaging system. HSI: VNIR HSI camera. (**b**) Photo of the hyperspectral imaging system.

**Figure 3 sensors-21-05595-f003:**
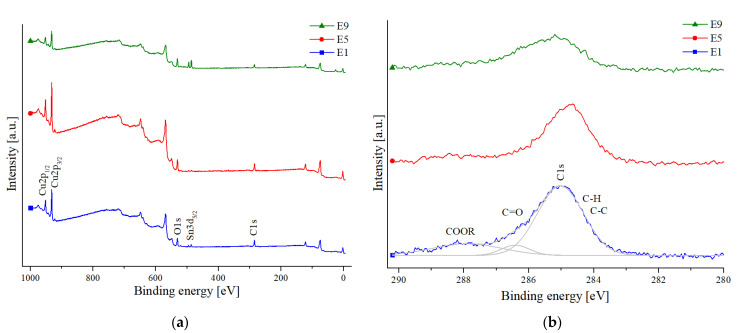
The XPS C1s and survey spectra of Set1, Set5, and Set7 measured with monochrome Al Kα radiation: (**a**) the XPS survey spectra; (**b**) the spectra C1s.

**Figure 4 sensors-21-05595-f004:**
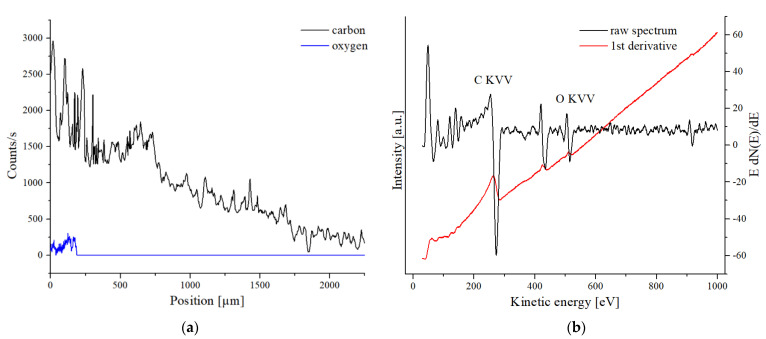
(**a**) The AES line scan of sample Set5 is shown. (**b**) A representative Auger spectrum of Set4 is shown. The carbon count rate in the line scan was collected from the C KVV Auger peak at 269.5 eV. The oxygen count rate was obtained from the O KVV peak at 514.5 eV.

**Figure 5 sensors-21-05595-f005:**
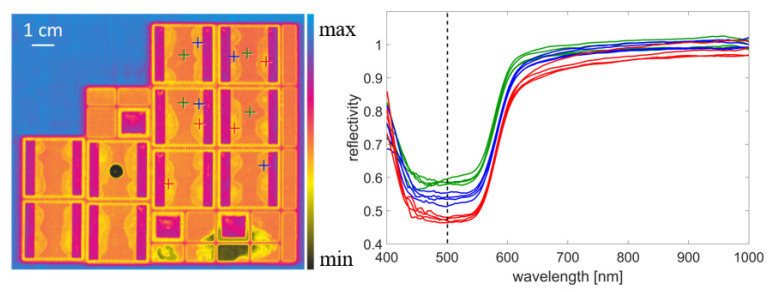
Result of the HSI measurement of the samples from Set2. Left: Color coded reflectivity at a wavelength of 500 nm. Right: Example spectra from different areas of the sample (clean, lightly contaminated, heavily contaminated). The position of the spectra is marked by colored crosses in the left image.

**Figure 6 sensors-21-05595-f006:**
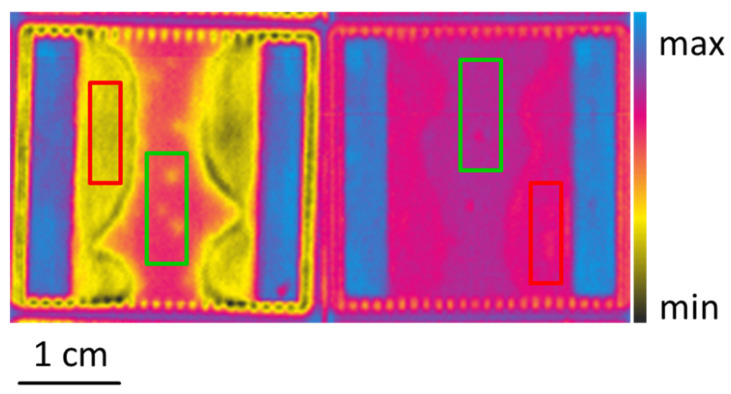
Example of the generation of training data from the HSI measurements of a sample from Set1 (left) and Set7 (right). The image shows the color-coded reflectivity at 600 nm. The red box marks Spot A and the green box marks Spot B.

**Figure 7 sensors-21-05595-f007:**
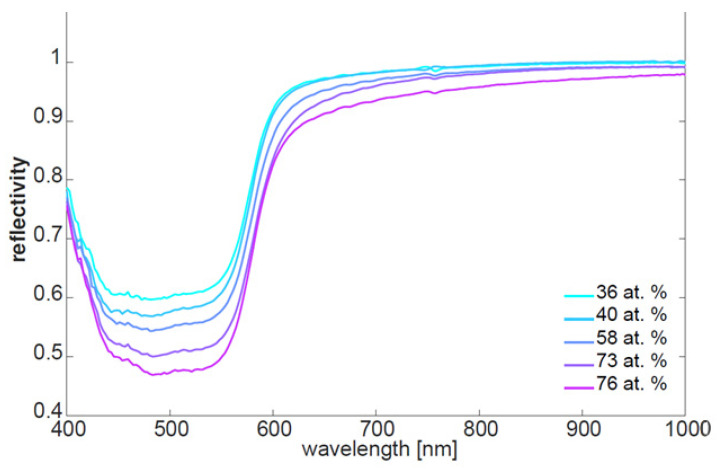
Mean spectra of the samples contaminated with different carbon content levels.

**Figure 8 sensors-21-05595-f008:**
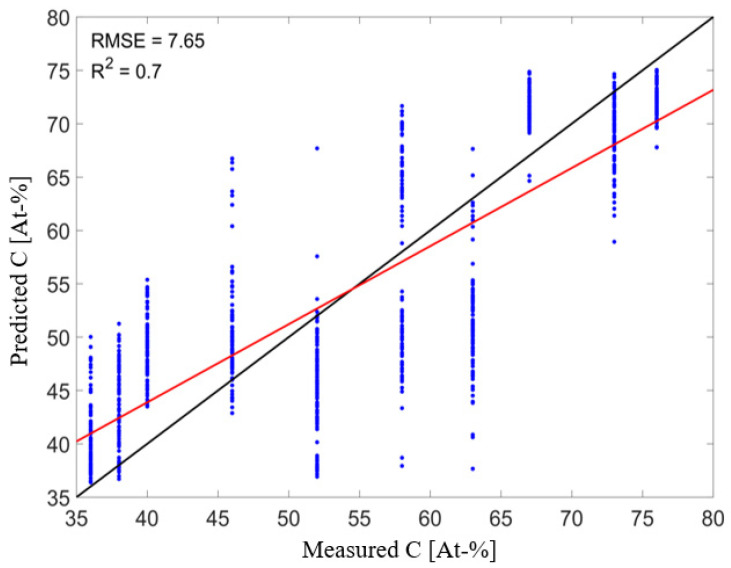
Prediction of the test data by the best-found RF regression model.

**Figure 9 sensors-21-05595-f009:**
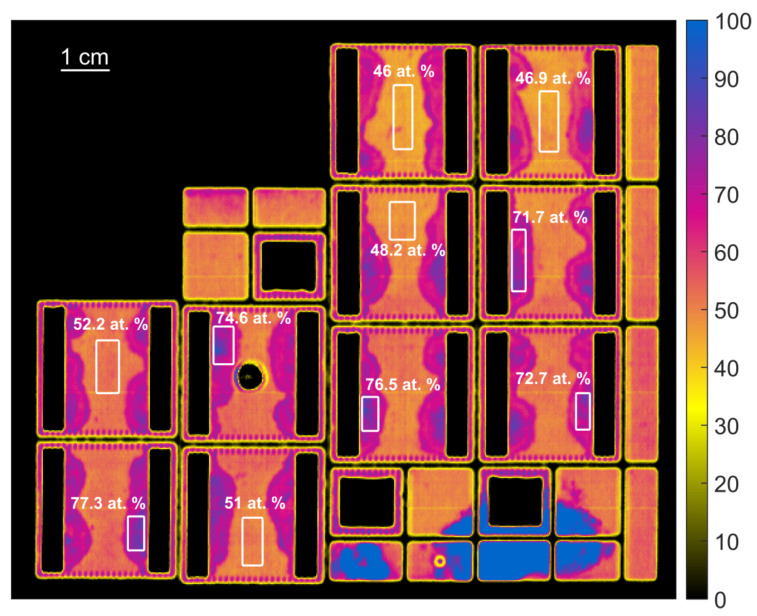
Prediction of the carbon content of the samples from Set2 with the best-found RF prediction model. The white areas were selected manually, and the values show the mean of the predicted carbon content. The reference XPS measurements for this samples were 73 At.-% (Spot A) and 58 At.-% (Spot B).

**Table 1 sensors-21-05595-t001:** The cleaning parameters for Set1–Set7 and the carbon content measured by XPS at Spot A and Spot B.

Set	Cleaning Temperature (°C)	Cleaning Duration (s)	A200 Conc. (%)	CI20 Conc. (%)	Rinsing Duration (s)	C Content Measured by XPS at Spot A (At.-%)	C Content Measured by XPS at Spot B (At.-%)
Set1	30	100	25	0	600	76	46
Set2	60	100	45	2	80	73	58
Set3	30	480	45	0	600	58	46
Set4	60	100	25	2	600	67	58
Set5	60	100	45	2	80	63	40
Set6	45	290	35	1	340	52	38
Set7	45	290	35	1	340	52	36

**Table 2 sensors-21-05595-t002:** Best found prediction RMSE and R2 for the prediction of the carbon content on the copper surfaces. There were three algorithms after hyperparameter optimization that were examined. The algorithms are shown in descending order, starting with the best-found one.

	RMSE [At.-% C]	R^2^
Random Forest	7.65	0.70
Elastic Net	7.85	0.68
Support Vector Machine	7.99	0.69

## Data Availability

The data presented in this study are available upon request from the corresponding author. The data are not publicly available due to privacy restrictions.

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
