# Peer review of "Use of Hyperspectral Imaging for the Quantification of Organic Contaminants on Copper Surfaces for Electronic Applications"

_sensors, 2021, doi:10.3390/s21165595_

Round 1

Reviewer 1 Report

the manuscript suggests the use of visible HSI systems to identify carbon content.

In general the idea is good but the manuscript needs to revise some aspects.

2.1. Sample preparation: add a photo describing the prepared samples.

2.5. Hyperspectral imaging (HSI): angle of incidence of light sources

3.2. HSI data evaluation and modeling. line 282: "Error! Reference source not found" It is probably an error in the writing of the manuscript.

To use the dataset in calibration in figure 6, did you choose processing to reduce the effects of light scattering? explain these aspects better.

Figure 7 shows a not very efficient RF regression model. Have you cross validated the data? My suggestion is to use a PLSDA or a non-linear classification method  with regression (PLS-KNN). In this way you can combine a regression method with a discriminating method and have more complete results on the possibility of using HSI-VIS in this case study. The results in Figure 8 confirm this hypothesis.

In addition the image in figure 8 requires an explanation with the data. Add a table with sensitivity and specificity in calibration cross validation and prediction of the result presented. 

Reviewer 2 Report

Thank you for inviting me to review this manuscript.

Some reflections:

  • The manuscript would benefit from English editing
  • The title does not reveal what processes are intended. It should be revised to better describe what is monitored
  • The abstract does not include any results. The results of the study, including specificity, sensitivity and statistical significance must be included in the abstract
  • The introduction is very long and could be shortened
  • 2. HSI data evaluation and modeling: Error! Reference source not found. Please fix this!
  • The authors state: “These fluctuations in the carbon contamination were observed by the AES measurement. In order to improve the accuracy of the models obtained, further XPS measurements of the carbon loading should be carried out and the HSI data for training should only be selected in the actual measurement range of the XPS measurements”. Why is the suggested improvements implemented in the experiments and repeated?
  • The discussion is very speculative and based on several hypotheses that are not proven.
  • The alleged correlation between HIS and carbon contamination is not clearly shown. Data on specificity, sensitivity and accuracy are lacking. A confusion matrix for the Random Forest model should be considered.

Reviewer 3 Report

The authors carefully revised the paper, and the quality has been grealy improved. I think the paper can be published after handling the following small issues:

  1. the title is too long and can be shorten.
  2. The limitation of this study should be mentioned with more details.

Round 2

Reviewer 1 Report

the reviewers edited the manuscript. as a methodological approach, it is acceptable. 

Author Response

Thank you.

Reviewer 2 Report

My comments are satisfactory addressed. My evaluation is in regard of HSI which is satisfactory described.

Author Response

Thank you.